# Autophagy Inhibitor Chloroquine Downmodulates Hepatic Stellate Cell Activation and Liver Damage in Bile-Duct-Ligated Mice

**DOI:** 10.3390/cells12071025

**Published:** 2023-03-27

**Authors:** Trinh Van Le, Hong-Thuy Phan-Thi, My-Xuan Huynh-Thi, Thanh Minh Dang, Ai Xuan Le Holterman, Gabriele Grassi, Thao-Uyen Nguyen-Luu, Nhung Hai Truong

**Affiliations:** 1Laboratory of Stem Cell Research and Application, University of Science, Ho Chi Minh City 700000, Vietnam; lvtrinh@hcmus.edu.vn (T.V.L.);; 2Viet Nam National University, Ho Chi Minh City 700000, Vietnam; 3Department of Pediatrics and Surgery, University of Illinois College of Medicine, Chicago, IL 60607, USA; 4Department of Life Sciences, University Hospital of Cattinara, University of Trieste, 34100 Trieste, Italy; 5Faculty of Biology and Biotechnology, University of Science, Ho Chi Minh City 700000, Vietnam

**Keywords:** anti-fibrosis, autophagy, bile duct ligation, chloroquine, stellate cell transformation

## Abstract

Hepatic stellate cell (HSC) activation via the autophagy pathway is a critical factor in liver fibrogenesis. This study tests the hypothesis that chloroquine (CQ) treatment can prevent autophagy and HSC activation in vitro and in vivo in bile-duct-ligated (BDL) mice. Sham-operated and BDL mice were treated with either PBS or CQ in two 60 mg/kg doses the day (D) before and after surgery. On day 2 (2D), HSCs were isolated, and their biological activities were evaluated by measuring intracellular lipid content, α-sma/collagen, and expression of autophagy lc3, sqstm1/p62 markers. The treatment efficacy on liver function was evaluated with serum albumin, transaminases (AST/ALT), and hepatic histology. Primary HSCs were treated in vitro for 24 h with CQ at 0, 2.5, 5, 10, 30, and 50 µM. Autophagy and HSC activation were assessed after 2D of treatment. CQ treatment improved serum AST/ALT, albumin, and bile duct proliferation in 2D BDL mice. This is associated with a suppression of HSC activation, shown by higher HSC lipid content and collagen I staining, along with the blockage of HSC autophagy indicated by an increase in p62 level and reduction in lc3 staining. CQ 5 µM inhibited autophagy in primary HSCs in vitro by increasing p62 and lc3 accumulation, thereby suppressing their in vitro activation. The autophagy inhibitor CQ reduced HSC activation in vitro and in vivo. CQ improved liver function and reduced liver injury in BDL mice.

## 1. Introduction

Autophagy is the process that transports defective intracellular materials for degradation and recycling in the lysosomal compartment [1,2]. Autophagy is an important regulator of hepatocyte homeostasis [3]. In the liver, its activity is upregulated in ischemia/reperfusion injury, alpha-1 antitrypsin deficiency, and alcohol or viral hepatitis [4].

The effectiveness of autophagy intervention in a model of liver disease has been questioned by studies. For instance, autophagy inhibition by chloroquine (CQ) reduced liver damage in the ischemia/reperfusion model of acute phage but worsened when treatment was prolonged [5]. CQ treatment was found to have positive effects in chronic [6] and acute [7,8] CCl_4_ models of liver failure; in addition, it prevented autoimmune hepatitis in humans [9] and inhibited liver cancer [10,11]. Additionally, the use of an autophagy inducer such as rapamycin mitigated liver damage in a model of nonalcoholic fatty liver disease [12] as well as in liver diseases of different etiologies [13]. Consequently, the purpose of this study was to examine the effect of an autophagy inhibitor in a model of cholestasis liver disease involving bile duct ligation (BDL).

Recent publications demonstrated that autophagy promotes hepatic stellate cell (HSC) activation [3], a critical step in hepatic fibrogenesis [14,15], and that inhibition of autophagy metabolic pathways can prevent HSC activation [16] and subsequent liver fibrosis [17]. HSC activation in fibrogenesis is characterized by the initiation stage and the perpetuation stage [18]. During the initiation phase, quiescent HSCs lose lipid droplet (LD) contents [16,19], acquire a contractile phenotype, and express α-SMA activation markers [14]. In the perpetuation phase, activated HSCs acquire a myofibroblast-like morphology with increased production of extracellular matrix (ECM) and express the proliferation marker Ki-67 [20]. The initiation phase is thought to be the most critical stage of HSC activation that determines subsequent HSC biological effects. In the CCl_4_-induced liver injury model, initiation occurs within one day of the CCl_4_ treatment [21]; in the partial hepatectomy [22] and BDL liver injury models, HSC initiation develops after 3 days and up to 9 days with over 80% of total cells activated [21]. In vitro, primary HSCs cultured on plastic dishes undergo initiation between 1 and 5 days [23,24] after plating. This study focuses on the acute phase of liver injury in the BDL model and on the early days of primary HSC in vitro culture, a condition that was not deeply investigated previously. This acute phase encompasses the initiation stage of HSC activation, when it may be more appropriate to intervene to prevent HSC activation.

CQ is an antimalarial drug and is considered an inhibitor of autophagic flux due to its ability to increase lysosomal pH and inhibit the activity of resident hydrolases [5]. Together, these effects suppress autophagic flexus at a late stage, resulting in an accumulation of autophagosomes in the cytoplasm without fusion with the lysosome, thus preventing the digestion of the packaged material [25]**.** The blocking of autophagic flexus through the accumulation of cytoplasmic autophagosome can be detected by its membrane compartment represented by p62/sequestosome 1 (sqstm1) [5,26].

Based on the above findings, we wanted to verify whether the autophagy inhibitor CQ may downregulate the activation of primary HSC in the initiation stage in vitro and in vivo, improving liver function in BDL mice with liver injury.

## 2. Materials and Methods

### 2.1. Reagents

The antibodies, enzymes, kits, buffers, and media (Dulbecco’s modified Eagle’s medium—DMEM, Fetal Bovine Serum—FBS, antibiotic-antimycotic solution, etc.) used in the study were purchased from Merck Millipore, Darmstadt, Germany, if not mentioned otherwise.

### 2.2. Ethical Declaration

The animal experiments were performed in accordance with the EU directive (2010/63/EU) and approved by the Animal Ethics Committee of the Stem Cell Institute, VNUHCM-University of Science, Vietnam (Ref N0: 200501/SCI-AEC).

### 2.3. Bile-Duct-Ligated Mice Model

The 3–4 month-old BALB/c mice were purchased from the Laboratory of Animal Use and Care (Stem Cell Institute, Ho Chi Minh city, Vietnam). Mice were maintained under 12 h light–dark cycle, 25 °C room, and fed ad libitum. BDL was performed under anesthesia with 10 mg/kg of Ilium xylazine-20 (Troy Laboratories, New South Wales, Australia) and 7 mg/kg of Zoletil (Virbac, Carros, France). Animals were treated for infection prophylaxis with 20 mg/kg, 2 doses/day of Lincomycin (Vemedim, Vietnam).

### 2.4. Experimental Design

A total of 35 male mice were used in 4 parts of the study, with 3 mice in each group.

Part 1.Investigation on the early time point of HSC activation on BDL including 0D (Sham), 2D, 7D, and 14D groups. BDL mice were treated with CQ (Cas: 50-63-5) 60 mg/kg or control 0.1 mL PBS (Phosphate buffer saline is CQ solution) i.p. the day before and the day after surgery and sacrificed at 2D for Part 2 and Part 3. The dose of CQ was referenced from previous studies in mice [5,26,27].Part 2.Investigation on the effects of CQ on BDL liver injury on 2D included Sham, BDL + PBS, and BDL + CQ groups.Part 3.Investigation on the effects of CQ (60 mg/kg) on HSC activation in 2D BDL mice. (HSCs were isolated from the liver of Sham, BDL + PBS, and BDL + CQ mice groups.)Part 4.In vitro effects of autophagy inhibitor CQ on primary HSCs.

### 2.5. HSC Isolation

In anesthetized mice, HSCs were isolated from the perfused liver using a two-step perfusion protocol with 0.05% pronase E (Cas: 9036-06-0) and 0.05% collagenase D (Cat: 11 088 858 001), followed by a mixture of pronase E, collagenase D, and 1% DnaseI (Cat: 260913) for 20 min to obtain a single-cell suspension. The liver cell population was filtered through a cell strainer, washed with Gey’s Balanced Salt Solution B, and separated by density gradient centrifugation in 9.6% Nycodenz (Cat 1002424, Axis-Shield, Dundee, UK) for 17 min at 1400 g. HSCs were harvested from the white layer after centrifugation [28].

### 2.6. In Vitro Autophagy Inhibition

Isolated HSCs were cultured at a density of 2.10^4^ cells/cm^2^ at 37 °C, and 5% CO_2_, in DMEM supplemented with 15% FBS, 1% antibiotics, for 24 h. Attached cells were treated with CQ at concentrations of 0, 2.5, 5, 10, 30, and 50 µM for 24 h. Autophagy activity was examined after two days of treatment.

### 2.7. CCK8 Assay

Cell proliferation was measured using CCK8 kit (Product no 96992) according to the manufacturer’s instructions. In brief, the CCK8 reagent was mixed with the culture medium to reach the working concentration of 10%. Then, cells were incubated with 100 µL of the working solution for 4 h followed by measuring the absorbance at 450 nm using a DTX-880 microplate reader (Beckman Coulter, Brea, CA, USA).

### 2.8. Oil Red O Staining (ORO)

Briefly, fixed cells were washed with isopropanol 60% for 5 min, stained with 0.5% ORO solution for 20 min, washed with isopropanol, and counterstained with hematoxylin for 30 s. The images of stained cells were captured using the inverted microscope, and quantification of the lipid droplet area was performed using ImageJ software.

### 2.9. Real-Time RT-PCR

Total RNA was extracted from 10 mg snap-frozen fresh liver tissue or isolated HSCs using the easy-BLUE™ Kit (iNtRON Biotechnology, Gyeonggi-do, Republic of Korea) and reverse transcribed into cDNA using the SensiFAST™ cDNA Kit (Bioline, London, UK). The cDNA was subjected to real-time PCR (SensiFAST™ Real-Time PCR Kits, Bioline). Semi-quantitative gene expression analysis was calculated using the Livak method of 2^−ΔΔCt^ [29]. The primer list is shown in Table 1.

### 2.10. Immunocytochemistry (ICC)

Attached cells were washed with PBS^-^ and fixed in 1% paraformaldehyde (PFA) for 30 min before staining. Fixed cells were permeabilized with 0.1% Triton X-100 for 10 min, blocked with blocking buffer (PBS, 4% Goat serum and 1% BSA/bovine serum albumin) for 30 min, and incubated with anti-desmin, anti-sqstm1/p62, anti-lc3b, and anti-collagenI primary antibodies at 4 °C overnight and secondary conjugated Alexa Flour 488 antibody for 1 h. DAPI mounting medium was added to stain the nucleus (Cas 28718-90-3, Santa Cruz, Dallas, TX, USA). Fluorescence images were captured using an Axio A1 Microscope (Carl-Zeiss, Oberkochen, Germany). Fluorescence intensity was quantified as the corrected total cell fluorescence (CTCF) using ImageJ software.

### 2.11. Serum Test

Peripheral blood was collected from the facial vein to measure serum albumin using the QuantiChrom™ BCG Albumin Assay Kit (Bioassay Systems, Hayward, CA, USA). Serum AST/ALT was measured using the GOT/GPT IFCC Mod. liquid UV Kit (Diagnosticum Zrt, Budapest, Hungary), and total leukocyte count using a hemocytometer.

### 2.12. Histopathology

The liver was perfused with PBS to wash out the blood and subsequently fixed in 4% PFA overnight. Fixed tissues were then dehydrated in sucrose 30%, embedded in paraffin, and sliced to 4 μm of thickness. Liver slices were deparaffinized with xylene and rehydrated with serial ethanol before staining.

### 2.13. Hematoxylin and Eosin (H&E)

H&E staining was conducted by standard protocol using Papanicolaou’s solution 1a Harris’ hematoxylin solution and Eosin Y-solution 0.5% aqueous solution. Histologic necrosis was measured as the percentage of necrotic area per tissue area using the Axio Vision software (Carl Zeiss AG).

### 2.14. Picrosirius Red

Slices were stained with the Picrosirius Red F3B Kit (Cat 24901-500, Poly Science, Niles, IL, USA), according to the manufacturer’s instructions. Liver fibrosis was quantified as the percentage of Picrosirius red positive area from 10 random lobules/slide by RGB stack mode of green channel (to highlight the red-stained collagen areas) in ImageJ software.

### 2.15. Immunohistochemistry (IHC)

Paraffin sections that had been deparaffinized and rehydrated underwent antigen retrieval using citrate buffer 10 mM pH 6 at 100 °C for 20 min, permeabilized with 0.25% Triton X-100 in PBS for 10 min, blocked with blocking buffer for 1 h. Slides were stained with primary antibody anti-α-sma or ck7 (cytokeratin 7) overnight at 4 °C. Cellular peroxidase blocking was conducted with peroxidase blocking buffer for 10 min at RT and stained with HRP-conjugated secondary antibody for 1.5 h. Immunocomplexes were developed using the AEC Chromogen Kit according to the given instructions. Cell nuclear was stained by hematoxylin. Slides were washed using tris buffer saline, 0.1% Tween-20 TBS (2,-3 times), between each step. The positive area was quantified as described in Picrosirius red staining.

### 2.16. Statistical Analysis

Data are presented as mean ± SD. Data were analyzed and graphed using GraphPad Prism software. Statistical analysis was conducted using ANOVA, and the differences between the groups were considered statistically significant if the *p*-value < 0.05.

## 3. Results

### 3.1. Liver Damage of Bile Duct Ligation

Common BDL-induced mice jaundice is visible in the ears, tail, and fingers (Figure 1A) due to the accumulation of bilirubin in peripheral blood. Macroscopically, the BDL liver was swollen and paled with an enlarged gallbladder compared to control animals (Figure 1B–E). At the microscopic level, the BDL liver showed inflammation, ductular reaction (a marker of bile duct proliferation), and areas of necrosis (Figure 1F–I,M). At the systemic level, BDL mice showed a dramatic increase in liver necrosis markers AST/ALT on day 2 after BDL (Figure 1J,K). AST/ALT remained elevated until day 7 and 14, although at a reduced level compared to day 2. The level of circulating albumin, a known marker of liver function, was significantly reduced at all time points; however, it was lower on day 7 compared to days 2 and 14 (Figure 1L). The extension of liver necrosis increased over time (Figure 1M); however, it did not perfectly overlap ALT/AST levels that, in contrast, tended to decrease on day 7 and 14 after BDL compared to day 2.

### 3.2. D2 after BDL as an Earlier Point of HSC Activation

We first analyzed the expression of collagen type I fiber, a hallmark of fibrogenesis, using Sirius red staining (Figure 2A–D). Collagen deposition in BDL mice increased over time (Figure 2I) along with α-sma protein and mRNA expression (*acta 2)* (Figure 2E–H,J,K), with day 2 being the earliest time point of the model. To confirm the expression of the activation marker in HSC, we isolated the cell from 2D BDL mice to analyze the gene expression of HSC activation markers such as *lrat* and *acta 2* (α-sma) (Figure 2L,M). As expected, the expression of these markers was significantly changed, with the increase of *acta 2* and the decrease of *lrat* compared to HSC from Sham mice (*p* < 0.01 and 0.05, respectively). Additionally, the *Map1lc3b* (lc3b) gene in HSC from 2D BDL mice was also increased compared to Sham mice, *p* < 0.001 (Figure 2N). Because we consider HSC initiation to be the most critical stage of HSC fibrogenic activities, we focus on the evaluation of CQ effects at the day 2 time point.

### 3.3. CQ Reduced D2 BDL-Induced Acute Liver Damage

The levels of circulating AST and ALT were significantly reduced in 2D BDL mice treated with CQ (BDL/CQ) (Figure 3Q,R) compared to control PBS treated animals (BDL/PBS). Albumin level was not affected because of the earliness of this time point (Figure 3S). CQ treatment was also associated with decreased systemic inflammation as shown by circulating leukocyte levels (Figure 3T) compared to control PBS-treated animals.

At the histological level, Sirius staining, α-sma, and bile duct proliferation (as shown by lower counts of ck7 positive ducts) were diminished in BDL-CQ liver (Figure 3A–D,I–L) without a significant change in the extent of liver necrosis (Figure 3E–H). These results support the concept of a beneficial effect of CQ on BDL liver damage.

### 3.4. Autophagy Inhibition by CQ Was Associated with HSC Inactivation in 2D BDL Mice

To evaluate whether the improved liver function in CQ-treated mice could be related to the inhibition of HSC activation, HSCs were isolated from Sham, D2 BDL + PBS, and BDL + CQ animals. Freshly isolated HSCs from Sham showed the typical characteristics of HSC in size (around 15 μm) and in the abundance of lipid droplets, as shown by both phase-contrast microscope and the intensity of ORO staining (Figure 4A,B). After plating, HSCs displayed their typical stellate structure and stained positively for desmin (Figure 4C), a well-known marker for HSC [30].

To evaluate autophagy, the cells were stained by sqstm1/p62, whose intracellular accumulation occurs during autophagy inhibition [31]. HSC from CQ-treated BDL liver showed increased levels of sqstm1/p62 (Figure 4D–F), consistent with the downmodulation of autophagy marker lc3 staining (Figure 4G–I), whose levels decrease upon autophagy inhibition.

The effect of CQ on HSC activation was also assessed by ORO staining, which revealed the amount of lipid droplets in the cells. Typically, in HSCs, lipid droplets decrease in parallel with their activation [30]. As expected, ORO staining that was visible in quiescent HSCs isolated from normal mice (Figure 5A) was reduced in HSCs isolated from BDL/PBS liver (Figure 5B). However, ORO staining was significantly increased in BDL mice treated with CQ (Figure 5C,D). As an additional marker of HSC activation, we evaluated the level of collagen I, known to be a profibrotic collagen. Consistent with the ORO staining finding, collagen expression was significantly decreased in BDL/CQ HSCs compared to the BDL/PBS control (Figure 5E–H). Because the mRNA levels of *acta 2* (*α*-sma) and *col1a1* (collagen type I) were not statistically different among the three groups of mice, CQ action may be limited to affecting protein stability, at least in the initiation stage of HSC activation (Figure 5I,J). Notably, we also did not detect significant differences in the mRNA levels for the proliferation marker *Mki-67*.

### 3.5. Confirmation of In Vitro Autophagy Inhibition and Activation in HSC Culture

To confirm the in vivo data, HSC isolated from non-treated animals were treated with CQ. Figure 6A depicts that the cytotoxicity of 30 µM CQ in HSC culture dramatically reduced HSC survival (*p* < 0.001). This observation was also confirmed by cell counting *(p* < 0.0001) (Figure 6B). In the CQ concentration range of 2.5–10 μM, cytotoxicity was still detectable and significantly reduced compared to a higher concentration. As we wanted to evaluate CQ effects on autophagy at a concentration that did not elicit a relevant HSC cytotoxicity, 5 µM CQ was considered. The 5 µM CQ inhibited autophagy, as demonstrated by the decreased levels of lc3 (Figure 6C–E) and the increased amount of p62 (Figure 6F–H) compared to the control (*p* < 0.05).

Figure 7 demonstrates the impact of CQ on HSC activation (2D). Specifically, compared with the control, CQ-treated HSCs prevented the loss of lipid droplets in cytoplasm in ORO staining (*p* < 0.05, Figure 7A–C) and reduced the expression of the activation marker collagen 1 (*p* < 0.05, Figure 7D–F). These results indicate that reducing autophagy flux with CQ also led to a reduction in the activation of HSCs in vitro.

## 4. Discussion

In liver fibrogenesis, ECM accumulates dramatically [14] following the activation of HSCs, portal fibroblasts, smooth muscle cells, and hepatocytes. Activated HSCs are the major fibrogenic cellular player, contributing to 82–96% of all MFBs (myofibroblasts) in toxic, cholestatic, and fatty liver diseases [32]. Therefore, downmodulation of HSC activation has great potential in reducing ECM deposition and thus the progression of liver fibrosis.

In this work, we explored the ability of CQ to downmodulate the early stage of HSC activation via the inhibition of autophagy using the BDL model of liver injury. Notably, we have not found studies that examine the initial stage of HSC activation in this model. The BDL model has typical characteristics of biliary obstruction disease (Figure 1A–E) and shares similar modifications of the biochemical markers (Figure 1J–M) reported in previous studies [33,34]. Additionally, the histological modifications (Figure 1F–I) unequivocally indicate the presence of portal reaction and inflammation, i.e., signs of liver injury. The molecular mechanisms responsible for the above observation and HSC activation in our BDL model may depend on several factors, including induced oxidative stress [35], metabolic products secreted by damaged liver cells [36,37], and increased synthesis of TGF-β due to inflammation [38]. Here, we combined the above reliable liver injury model with our expertise in the isolation of the HSCs [28]. The latter is not a trivial aspect as optimal HSC isolation and culturing conditions are essential to maintain the original phenotype of this cell type when put into culture.

This study focused on day 2 of BDL injury because this time point corresponds to the initiation stage of HSC activation. An early intervention is more likely to enable reversible changes in the liver fibrosis [39] before less reversible extensive cell damage and inflammation are established in the advanced injury stages [40]. Our data demonstrated the downregulation of autophagic flexus in HSCs obtained from the livers of CQ-treated animals. In particular, we have shown an increase in sqstm/p62 staining and a decrease in lc3 staining (Figure 4). In parallel, we observed the downregulation of HSC activation, shown by the increase of lipid droplets and the reduction of collagen I production and of α-sma (Figure 5). These findings were associated with decreased inflammation, reduced liver fibrosis assessed by α-sma expression, and reduced ductal proliferation (Figure 3). The phenomenon correlates with liver fibrosis and fibrosis progression [41]. While the inhibition of HSC autophagy by CQ improves BDL fibrosis via the mechanism we reported here, alternative CQ-based mechanisms are also under investigation in our lab.

There are multiple molecular mechanisms responsible for the activation of HSC autophagy, including (1) lipopolysaccharide-induced autophagy through its receptor TLR4, which activated the AKT-mTOR and AMPK-ULK1 pathways thus promoting autophagy [27]; (2) oxidative stress aggregated protein prime autophagy [35]; and (3) inflammatory cytokines such as TGF-beta 1- and IGF-induced autophagy through ERK-JNK [38] and PI3k/Akt/mTOR [42] signaling. The role of autophagy in HSC activation was due to LD digestion in the cytoplasm by a process termed lipophagy; this process supplies cell energy for cell transformation [19], for maintaining MFB proliferation, and for producing ECM. Therefore, the inhibition of autophagy can effectively downregulate HSC activation, eventually inducing cell apoptosis [43] and reducing liver fibrosis [17]. Consistent with these results, our study shows that autophagy inhibition in HSC preserves liver function and reduces liver damage.

CQ also had positive effects at the organ level as it reduced the markers of liver necrosis ALT and AST (Figure 3R,S). However, CQ could not reduce the necrotic areas (Figure 3E–H). In this regard, a previous paper [7] reported necrosis reduction following CQ administration in mice carbon tetrachloride-induced acute liver injuries. In contrast, in a rat liver ischemia/reperfusion injury model [5], the worsening of necrosis was reported 24 h after reperfusion. Thus, the effect of CQ on liver necrosis remains unclear and may be related to the type of animal used and/or the liver injury model employed.

CQ did not significantly improve the plasmatic level of albumin (Figure 3T), a well-known marker of liver function. In this regard, it should be noted that the decrease in BDL/PBS animals compared to normal animals is rather modest, being about 25% (Figure 3T). Thus, it is possible that in our model this marker cannot reflect a deficiency in liver function, at least at the time point considered. Regardless of these considerations, CQ has a beneficial effect at the systemic level as it can decrease the number of circulating leukocytes (Figure 3U), i.e., it can reduce systemic inflammation. The anti-fibrotic value of CQ is not limited to our BDL model. Indeed, this property was observed in other chemical-induced liver-injured models [6,16,17].

In primary HSCs, after 1 day of isolation and treatment with CQ at concentrations of 5 µM for 24 h, the expression of lc3 and p62 in the HSC cell population was decreased and increased, respectively, compared to untreated control cells (Figure 6). These results paralleled with the reduction of HSC activation after 2 days of treatment. Our findings are consistent with Hernandez et al. (2012), who found that CQ decreased lc3 and increased p62 due to the non-degradable accumulation of autophagosome bodies; however, in this study, autophagy was performed and observed in an immortalized activated HSC cell line (JS1) after 12 h of treatment with 10 µM CQ [17]. Our data obtained in a more relevant in vitro model (primary HSC) strengthen this concept, fully confirming it.

Autophagy can be inhibited pharmacologically in different liver cell types. How-ever, to target the effect on HSCs, the major players in liver fibrosis, cell targeted delivery systems have to be considered. By this approach, it is in principle possible to improve the agent’s efficacy and to reduce its side effects. The identification of an HSC-specific delivery system may not be too complicated to develop, as we have recently shown for the targeting of another type of liver cells, i.e., hepatocytes [44].

In conclusion, our findings demonstrate CQ’s therapeutic potential in HSC function and in improving liver injury outcomes. 

## 5. Conclusions 

CQ, an autophagy inhibitor, decreased HSC activation in vitro and in BDL mice, improving liver function and decreasing liver fibrosis and injury.

## Figures and Tables

**Figure 1 cells-12-01025-f001:**
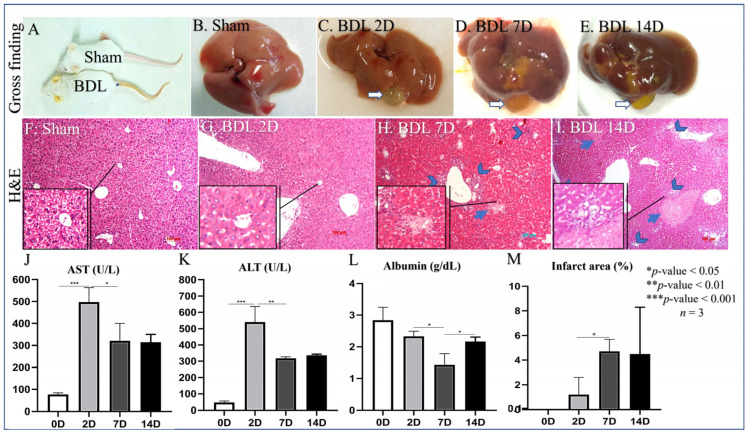
Evidence of liver damage in BDL mice. (**A**) Macroscopic evidence of jaundice in BDL mice compared to control animals. (**B**) Liver macroscopic appearance in control mice. (**C**–**E**) Liver macroscopic appearance in BDL mice 2, 7, and 14 days after BDL. (**F**) Microscopic appearance of liver in control animals (H&E staining). (**G**–**I**) Microscopic appearance of liver in BDL animals (H&E staining) 2, 7, and 14 days after BDL; head arrows indicate portal reaction and inflammation; arrows indicate the necrotic area. (**J**–**L**) Levels of circulating AST/ALT/Albumin in BDL mice 0, 2, 7, and 14 days after BDL, * *p* < 0.05, ** *p* < 0.01, *** *p* < 0.001, *n* = 3. (**M**) Quantification of necrotic areas in BDL mice Sham (0), 2, 7, and 14 days after BDL, * *p* < 0.05, *n* = 3, 100 µm scale bare.

**Figure 2 cells-12-01025-f002:**
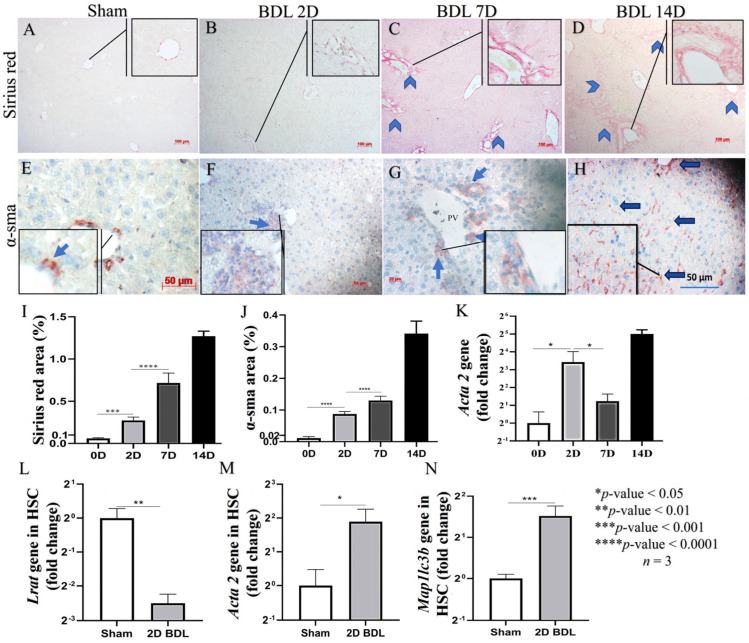
Hepatic stellate cell activation following bile duct ligation. (**A**) Sirius red staining of liver tissues in control animal. (**B**–**D**) Sirius red staining of liver tissues in BDL animals over time; head arrows indicate portal fibrosis. (**E**) α-sma staining of liver tissue in control animal. (**F**–**H**) α-sma staining of liver tissue in BDL animals over time; blue arrows indicate interstitial HSC and portal MBF positive for α-sma. (**I**) Quantification of Sirius red staining of liver tissues in BDL animals over time. (**J**) Quantification of α-sma in liver tissues in BDL animals over time. (**K**) Quantification of *acta 2* mRNA level in liver tissues of BDL animals over time. (**L**–**N**) mRNA levels of *lrat* (**L**), *acta* 2 (**M**) and *Map1lc3b* (**N**) in HSC isolated from Sham and BDL mice. * *p* < 0.05, ** *p* < 0.01, *** *p* < 0.001, **** *p* < 0.0001, *n* = 3, scale bar: 100 µm for sirius red and 50 µm for α-sma staning.

**Figure 3 cells-12-01025-f003:**
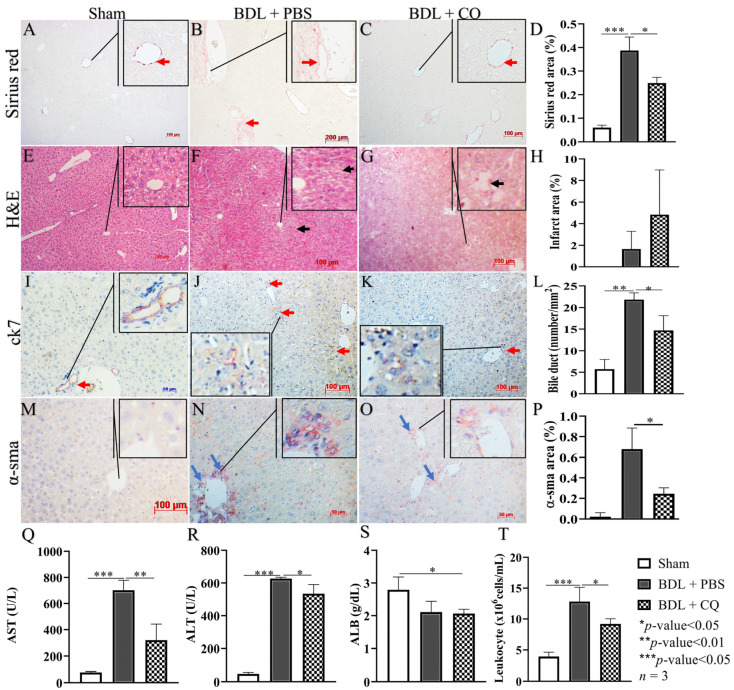
Effects of CQ treatment effects on liver damage in BDL. (**A**–**D**) Representative Sirius red staining of liver tissues in Sham, BDL + PBS, and BDL + CQ animals with the respective quantification; red arrows indicate positive areas. (**E**–**H**) Representative H&E staining of liver tissues in Sham, BDL + PBS, and BDL + CQ animals with the respective quantification; black arrows indicate areas of necrosis. (**I**–**L**) Representative ck7 staining in liver tissues in Sham, BDL + PBS, and BDL + CQ animals with the respective quantification; red arrows indicate portal reaction due to bile duct proliferation. (**M**–**P**) Representative α-sma staining in liver tissues from Sham, BDL + PBS, and BDL + CQ animals together with the quantification; blue arrows indicate positive areas. (**Q**–**S**) ALT, AST, and albumin levels in Sham animals and BDL (2 days) animals treated either with CQ (BDL/CQ) or with PBS (BDL/PBS, control). (**T**) Peripheral leukocytes in Sham animals and BDL (2 days) animals treated either with CQ or with PBS (control). * *p* < 0.05, ** *p* < 0.01, *** *p* < 0.001, *n* = 3, 100 µm scale bar.

**Figure 4 cells-12-01025-f004:**
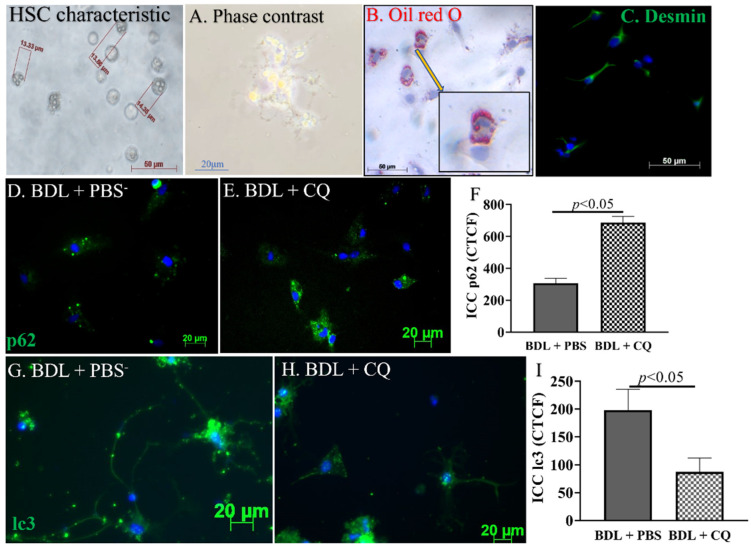
Effects of CQ on HSCs in vivo autophagy. (**A**–**C**) Representative images of the phenotypic characteristics of HSCs isolated from Sham mice: phase contrast images of cells 24 h after seeding (**A**), Oil red O staining (B), and desmin staining (C; green, desmin staining; blue, nucleus staining). (**D**–**F**) Representative images of the sqstm1/p62 staining in isolated HSCs and graph comparison of CTCF (Correct Total Cell Fluorescence quantification) from ICC staining. (**G**–**I**) Representative images of the lc-3b staining in isolated HSC and graph comparison of CTCF from ICC staining in BDL + PBS and BDL + CQ mice, *n* = 3.

**Figure 5 cells-12-01025-f005:**
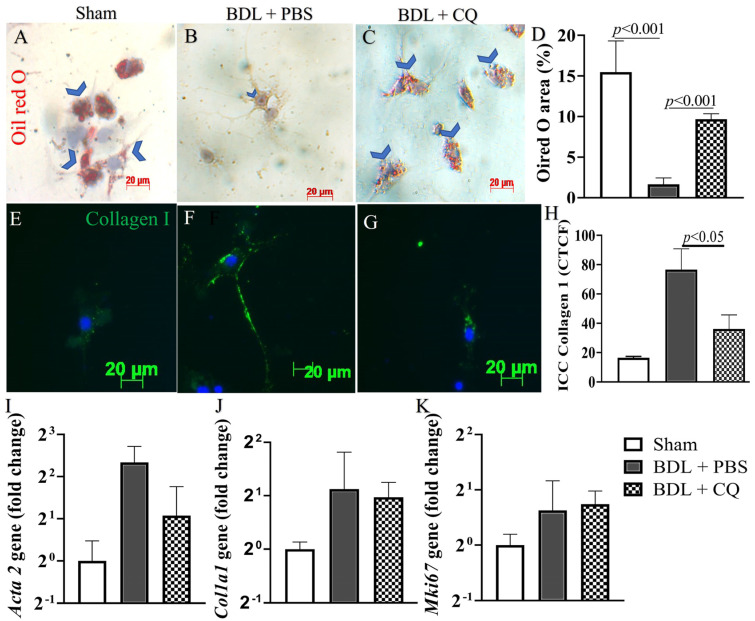
Effects of CQ on HSC in vivo activation. (**A**–**C**) Representative images of Oil red O staining in HSC isolated from the three mice groups: Sham, BDL/PBS, and BDL/CQ; blue arrows indicate the positive areas. (**D**) Quantification of the surface of Oil red O staining areas, *p* < 0.05, *n* = 3. (**E**–**G**) Representative images of type I collagen staining in HSCs isolated from the three mice groups: Sham, BDL/PBS, and BDL/CQ; (**H**) Quantification CTCF (Correct Total Cell Fluorescence quantification) of Collagen staining, *p* < 0.05, *n* = 3. (**I**–**K**) mRNA levels of *acta 2*, *col1a1,* and *Mki-67* markers in HSCs isolated from Sham, BDL/PBS, and BDL/CQ mice.

**Figure 6 cells-12-01025-f006:**
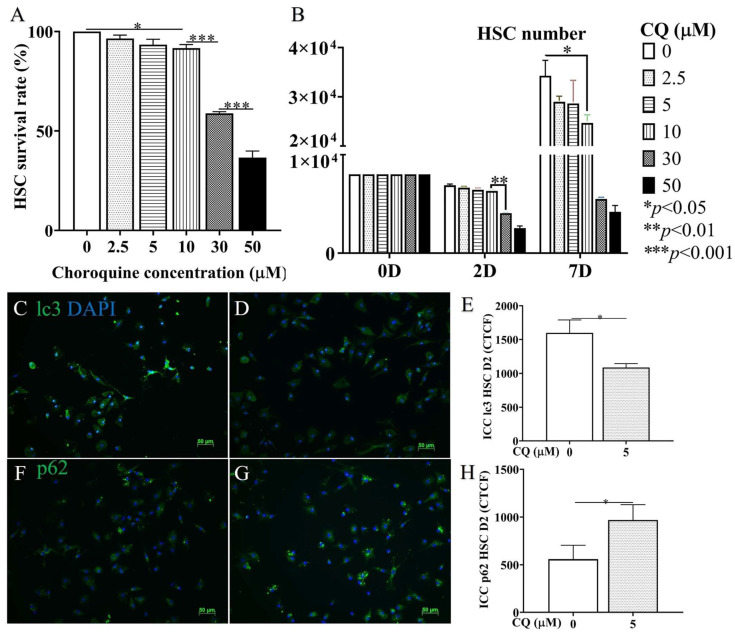
CQ cytotoxicity and autophagy effectiveness on HSCs in vitro. (**A**,**B**) Graphic comparison of CQ concentrations varying between 0, 2.5, 5, 10, 30, and 50 µM based on cytotoxicity and proliferation until 7D of culture (**B**) and 2D (**A**). (**C**,**D**) Representative images of ICC staining with lc3 antibody/DAPI of in vitro HSCs treated with 0 and 5 µM of CQ. (**E**) Graphic comparison of CTCF (Correct Total Cell Fluorescence quantification) from ICC staining. (**F**,**G**) Representative images of ICC staining with p62 antibody/DAPI of HSC treated with 0 and 5 µM of CQ. (**H**) Graphic comparison of CTCF from ICC staining. *p*-value: * < 0.05, ** < 0.01, *** < 0.001, *n* = 3, 50 µm scale bar.

**Figure 7 cells-12-01025-f007:**
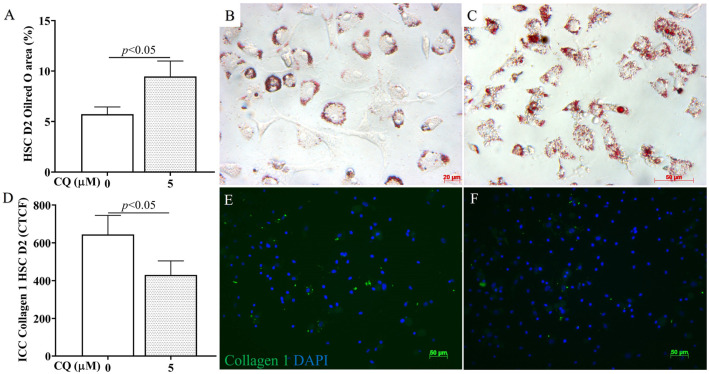
CQ effects on HSC in vitro activation. (**A**) Graphic comparison of Oil red O area (%) between CQ 0 and 5 µM from the ORO staining image. (**B**–**D**) Graphic comparison of CTCF (Correct Total Cell Fluorescence quantification) from ICC staining with collagen 1 antibody/DAPI (**E**,**F**) between CQ 0 and 5 µM. *p*-value < 0.05, *n* = 3, 50 µm scale bar.

**Table 1 cells-12-01025-t001:** Primer sequence and antibody information.

Primer	Sequence		Gene ID
acta2	F: GCATCCACGAAACCACCTA		NM_007392.3
	R: CACGAGTAACAAATCAAAGC		
col1a1	F: CAATGGCACGGCTGTGTGCG		NM_007742.4
	R: AGCACTCGCCCTCCCGTCTT		
gapdh	F: AAGTTGTCATGGATGACC		NM_001289726.1
	R: TCACCATCTTCCAGGAGC		
lrat	F: CTGACCAATGACAAGGAACGCACTC		NM_023624.4
	R: CTAATCCCAAGACAGCCGAAGCAAGAC		
Map1lc3b	F: TTCTTCCTCCTGGTGAATGG		NM_026160.5
	R: GTGGGTGCCTACGTTCTCAT		
Mki67	F: AATCCAACTCAAGTAAACGGGG R: TTGGCTTGCTTCCATCCTCA		NM_001081117.2
**Antibody**		**Source**	**Dilution**
collagen I		Abcam, ab270993	1:200
cytokeratin 7		Abcam, ab199718	1:500
desmin		Abcam, ab8592	1:300
lc-3b		Abcam, ab192890	1:200
α-sma		Abcam, ab15734	1:300
sqstm1/p62		Abcam, ab91526	1:500
Goat Anti-Rabbit IgG H&L (Alexa Flour 488- conjugated)		Abcam, Ab150077	1:500
Goat anti-Rabbit IgG HRP-conjugated		Abcam, ab6721	1:500

## Data Availability

Data is contained within this article.

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
