# Peer review of "Autophagy Inhibitor Chloroquine Downmodulates Hepatic Stellate Cell Activation and Liver Damage in Bile-Duct-Ligated Mice"

_cells, 2023, doi:10.3390/cells12071025_

Round 1

Reviewer 1 Report

In this investigation, the authors show interesting evidence on the effect of cloroquine in the activation of HSC in both in vivo and in vitro models. However, before further steps in the publication process, some issues still need to be addressed.

Genes symbols should be defined at the first appearance in the manuscript, subsequently, only the symbol can be cited. In page 5, authors cite lrat and sma-α, and lc-3b genes but they are not defined. Moreover, sma-α is not the symbol for alfa smooth muscle actin gene, and sometimes it is cited as sma-α but in others is cited as α-SMA. Based on HUGO Gene Nomenclature Committee, gene and protein symbols must be indicated according to the accepted nomenclatures. So, throughout the review manuscript including text, tables, figures and figure legends, gene and protein symbols must be properly indicated. Authors should know that both gene and protein short names are symbols, not abbreviations, and their symbols are different among species. As reference, authors should also read the following article: PMID: 22836666, as well as, GeneCards website.

In the following sentence authors stated “Reagents: antibodies, enzymes, kits, buffers, and media (DMEM, FBS, antibiotic antimycotic solution, etc.) in the study were purchased from Sigma Aldrich, USA if not mentioned”. However, for reproducibility of the results by others researchers, catalogue number, vendor, place, etc., of all reagents should be included.

In page 5 authors indicated the following sentence “Table 1. Primer sequence and antibody information”, however, they did not include the table. Primers sequence of genes should include the NCBI accession number.

Authors pointed out that quantitative gene expression analysis was calculated by the Livak method of 2-ΔΔCt, however they did not cite Livak´s reference.

In the following sentence, what does “D” stand for? “Part 1: Investigation on the early time point of HSC activation on BDL including 0D (Sham), 2D, 7D, 14D groups”. Abbreviations and symbols should be defined at the first appearance in the manuscript.

In some cases, the symbol indicating Celsius degrees is not properly used, instead, authors included “0” as superscript.

Why authors did not include a zoom frame for figure 2 panel G, as included in panels E, F and H?

Pictures of figure 3 should be substituted by others of bigger magnification since the indication by arrows are not clearly visualized, moreover, arrows are too big.

Reviewer 2 Report

The present study reported on autophagy inhibition by Chloroquine in vitro and in vivo.

Please find the comments below:

1.    What is the novelty of the current work aside from the model used, in view many of the published literatures have highlighted on the mechanism underlying autophagy inhibition by CQ? The biomarkers analysed in this current work also similar to what have been published.

2. How was the dose CQ 60 mg/kg chosen? How do this dose correspond to humans, taking into account the differences in metabolism between mice and humans?

3. The methods should be described in further detail. For example:

            I.        How many days were the mice acclimatized prior to the start of the in vivo study? Are the mice male or female?

           II.        How was CQ prepared for the experiment, i.e in solution or suspension in saline/water?

         III.        In vitro autophagy inhibition- what is the seeding density of the HSC? Why and how 0, 2.5, 5, 10, 30, and 50 µM chosen?

          IV.        What is the amount of fresh liver tissue or isolated HSCs used for the RT-PCR?

           V.        Does the methods in current work follows previously published literatures since references are missing.

4. There is no coherence between the methods for in vitro and the results. In methods, it was mentioned that the cells were cultured and examined after 2 days of treatment while the results are showing the outcome until the 7th day of culture.

5. Why was 5 µM CQ chosen, not 10 uM for further assay (in-vitro)?

6. The discussion is somehow repeating the results and is not in-depth enough. This part need to be improved by incorporating more current references to enrich this work.

7. Suggest incorporating suggestions for future work, and future perspectives (if any).

Reviewer 3 Report

This study reported that autophagy inhibitor Chloroquine (CQ) improved liver function and reduced liver injury in BDL mice via autophagy inhibition and activation modulation on HSCs. There were some issues should be addressed as below.

1. Le TV, Dinh NBT, Dang MT, Phan NCL, Dang LTT, Grassi G, Holterman AXL, Le HM, Truong NH. Effects of autophagy inhibition by chloroquine on hepatic stellate cell activation in CCl4-induced acute liver injury mouse model. J Gastroenterol Hepatol. 2022 Jan;37(1):216-224. doi: 10.1111/jgh.15726. Epub 2021 Nov 9. PMID: 34713488.

Apart from the way in which liver fibrosis is induced in mouse models, what is the difference of the current work?

2. In the Introduction section, please introduce the shortcomings of current domestic and foreign research and explain the innovation of this research.

3. Section 3.1, what the comparison of Fig.1 J/K/M illustrates?

4. In the section of results 3.2, the expression of Irat gene in D2 BDL mice in Fig.2L is lower than that of mice in the sham group, which is not consistent with the description of the results in Line 187.

5. The quantification of hepatic necrotic areas in Fig.3H is not consistent with Fig.3F-3G.

6. Fig.6C-E shows increased LC3 expression in 5 µM CQ-treated HSC, contradicting the previous article that LC3 expression was decreased upon inhibition of autophagy. Please confirm if the experimental data are incorrect.

7. Line 313 of the Discussion section, please confirm how the lipid droplets in HSC change when HSC activation is downregulated.

Reviewer 4 Report

Please revise english and references

Reviewer 5 Report

This study investigated whether the autophagy inhibitor Chloroquine (CQ) can prevent autophagy and HSCs activation in vitro and in vivo in bile-duct ligated mice. The results showed that CQ treatment improved liver function and reduced liver injury in BDL mice by suppressing HSCs activation, as indicated by higher HSC lipid content and collagen I staining. Furthermore, the study also found that CQ inhibited autophagy in primary HSCs in vitro. These findings suggest that CQ could be a potential therapeutic approach for liver disease. The study has potential but needs improvement to be published in this journal. Please find my suggestions and comments below-

Comments:

1) The authors have used 60 mg/kg of CQ for their experiment. What was the reasoning for administering 60 mg/kg of CQ, and how did the author establish this dosage? What is the time duration of the dose?

2) In methods & material, the authors have used CCK8 assay. Where is the data from these experiments?

3) In Fig1 F-I, Provide images at better magnification so that readers can appreciate the clinical features. Please zoom in on the image in the corner of every image. 

4) In Fig1 J-K, there is an increase in AST and ALT levels on Day-2. Then, AST and ALT levels stabilized on Day-7 and Day-14. Is there any clinical significance? 

5) Please be consistent with the figure preparation. Correct the image Fig2 G, same as Fig2 E, F, and H.

6) In the text, the authors mention that lrat genes increase on Day-2. However, we did not see increased mRNA expression in Fig 2D. Correct the text.

7) Consistency in labeling is required across all figures, including the use of consistent symbols and labeling for the x-axis and y-axis.

8) In Fig3, Provide images at better magnification so that readers can appreciate the clinical features. Please zoom in on the image in the corner of every image.

9) In figure 4, Please provide mRNA and western blot data for p62 and LC3B to support your finding. BDL+ PBS and BDL +CQ?

10) What does PBS- indicate in Figures 3, 4, and 5? Does it imply that PBS- was not administered?

11) The legends of the figures do not offer specific information about the figures. However, both the figures and their legends should be understandable without additional explanation. Please work on them.

12) Page 9, line 267, please provide support for this statement. How autophagy is inhibited, and LC3B is increasing. Explain?

13) Provide the information from where all the materials and antibodies are procured for the experiments.

14) In vivo and In vitro should be in italics.

15) The changes in p62 levels depend on the context and specific cellular 

process involved in autophagy p62 levels, while particular liver pathological diseases like NAFLD are typically decreased. Researchers have seen an increase in p62 in impaired autophagy with a lot of lipid accumulation. Please highlight the phenomena in your model.

16) Graphical model is necessary, which will explain this manuscript. 

Round 2

Reviewer 1 Report

Although authors have properly addressed most of suggestions, they partially addressed the proper rules to cite gene symbols; for example, “acta2” must be written as “Acta2” in italics. In HUGO Gene Nomenclature Committee website authors can find the details…

All gene symbols in Table 1 need to be corrected as well as in the whole manuscript.

Reviewer 2 Report

Authors have sufficiently addressed most of the suggestions.

Reviewer 3 Report

The contradicting results were explained or revised.